# Insomnia symptoms and neurofunctional correlates among adults receiving buprenorphine for opioid use disorder

**Augustus M. White**[1]*, **Michelle Eglovitch**[1], **Anna Beth Parlier-Ahmad**[1], **Joseph M. Dzierzewski**[2], **Morgan James**[3], **James M. Bjork**[1], **F. Gerard Moeller**[1], **Caitlin E. Martin**[1]

**1** School of Medicine, Virginia Commonwealth University, Richmond, Virginia, United States of America,
**2** National Sleep Foundation, Washington, DC, United States of America, **3** Department of Psychiatry, Rutgers University, Newark, New Jersey, United States of America

☯ These authors contributed equally to this work.
\* whiteam25@vcu.edu

## Abstract

### Objectives

Insomnia symptoms are negatively related to opioid use disorder (OUD) treatment outcomes, possibly reflecting the influence of sleep on neurofunctional domains implicated in addiction. Moreover, the intersection between OUD recovery and sleep represents an area well-suited for the development of novel, personalized treatment strategies. This study assessed the prevalence of clinically significant insomnia symptoms and characterized its neurofunctional correlates among a clinical sample of adults with OUD receiving buprenorphine.

### Methods

Adults (N = 129) receiving buprenorphine for OUD from an outpatient clinic participated in a cross-sectional survey. Participants completed an abbreviated version of NIDA's Phenotyping Assessment Battery, which assessed 6 neurofunctional domains: sleep, negative emotionality, metacognition, interoception, cognition, and reward. Bivariate descriptive statistics compared those with evidence of clinically significant insomnia symptoms (Insomnia Severity Index [ISI] score of ≥11) to those with minimal evidence of clinically significant insomnia symptoms (ISI score of ≤10) across each of the neurofunctional domains.

### Results

Roughly 60% of participants reported clinically significant insomnia symptoms (ISI score of ≥11). Experiencing clinically significant insomnia symptoms was associated with reporting greater levels of depression, anxiety, post-traumatic stress, stress intolerance, unhelpful metacognition, and interoceptive awareness (ps<0.05). Participants with evidence of clinically significant insomnia were more likely to report that poor sleep was interfering with their OUD treatment and that improved sleep would assist with their treatment (ps<0.05).

**Data Availability Statement:** All data files are available via Open Science Framework Project. These files are currently available to review.

Interested researchers can access the raw (deidentified) dataset, analytic code, and log file from the following: White A. Project SORT - Insomnia, Neurofunction, and Sleep - PLOS ONE 2023. osf.io/fgc3t.

**Funding:** Clinical and Translational Science Award No. UM1TR004360 from the National Center for Advancing Translational Sciences, National Institute on Drug Abuse (NIDA) award no. K23DA053507 (PI: Martin), NIDA award no. T32DA007027 (PI: W. Dewey), NIDA award no. F30DA057047 (PI: White), NIDA award no. R00DA045765 (PI: James), and NIDA award no. UG1DA050207 (PI: Moeller) from the National Institutes of Health. Findings do not necessarily reflect the viewpoints of the funders. The funders had no role in study design, data collection and analysis, decision to publish, or preparation of the manuscript.

**Competing interests:** The authors have declared that no competing interests exist.

## Conclusions

Insomnia was prevalent among adults receiving buprenorphine for OUD. Insomnia was associated with neurofunctional performance, which may impact OUD treatment trajectories. Our findings indicate potential targets in the development of personalized treatment plans for patients with co-morbid insomnia and OUD. To inform the development of novel treatment strategies, more research is needed to understand the potential mechanistic links between sleep disturbances and substance use.

## Introduction

Both opioid intoxication and withdrawal disrupt sleep architecture at a physiological level [1, 2]. At the same time, sleep disturbances themselves predispose individuals to opioid use disorder (OUD) [3–5]. A growing body of literature indicates that *neurobiological*, *neuropsychiatric*, and *social-ecologic* factors might mediate the bi-directional relationship between sleep disturbances and OUD [4, 6]. These are important relationships to understand because sleep disturbances (e.g., insomnia) are common among individuals receiving medication for OUD (MOUD), like buprenorphine [7], and are associated with negative OUD treatment outcomes, including substance use recurrence. Thus, strategies to improve sleep may be a novel approach for improving treatment outcomes for people with OUD [6, 8].

Previous evidence indicates that poor sleep contributes to the development and maintenance of substance use disorders by heightening pain sensitivity, promoting negative affect, and adversely impacting stress reactivity as well as self-care-related executive functions [1, 4, 9–11]. These relationships may arise via a depletion of "cognitive resources" [12] or disruption of neurobiological systems important for arousal, such as imbalance in the orexin/hypocretin system [3, 13, 14]. In humans, these disruptions can manifest as differences in incentive-related decision-making that can be measured using validated metrics [9, 13, 15–17]. Moreover, sleep quality can impact how individuals choose to respond (e.g., to avoid) exertion of effort [16], which could be a mechanism for poorer performance on a variety of tasks. Sleep quality may thus influence a set of neurofunctional domains that are believed to predict substance use treatment outcomes [21]. However, previous studies have typically considered only a narrow set of neurofunctional indices within the context of a single design or relied on samples whose subjects were actively engaged in substance use. Thus, the relationship between sleep and broad neurofunctional outcomes in addiction treatment samples is not well characterized.

Here, we attempt to bolster the existing literature by examining the prevalence of insomnia and its neurofunctional correlates in the domains of reward, cognition, interoception, negative emotionality, and metacognition among a sample of individuals receiving buprenorphine for OUD. Sleep disturbances such as insomnia may alter the underlying neurofunctional mechanisms of OUD at various stages of the OUD treatment and recovery. Obtaining insights on how sleep disturbances relate to other symptomatology would be useful in the creation of individualized treatment plans for patients with OUD and comorbid insomnia by identifying potential targets for intervention [9] such as adjunctive pharmacologic (e.g., orexin receptor antagonists, antidepressants, etc.) and non-pharmacologic (e.g., CBT-I, working memory training, cue exposure therapy, etc.) treatments to medications for OUD. Given the multi-factorial (e.g., genetic, environmental, lifestyle) etiology of both sleep disturbances and substance

use behaviors [18], the capacity for personalization in treatment for this population represents a promising avenue to improve OUD outcomes [1, 19, 20].

## Materials and methods

Recruitment and testing procedures of this protocol were approved by the Virginia Commonwealth University Institutional Review Board.

### Sample

We recruited individuals aged 18–65 who met criteria for OUD and had been receiving buprenorphine for at least 6 weeks from an outpatient substance use clinic in Virginia, USA. We excluded patients with a serious comorbid cognitive or psychiatric impairment (as determined based on chart review, clinical impression of the research assistant, or an inability to complete informed consent procedures), with a language barrier, or women who were pregnant or within 6 weeks of the end of a pregnancy. Enrollment occurred from 14 February 2022 through 15 September 2023. We screened a total of 350 individuals and 209 consented to participate. From the group of consenting individuals, we included the N = 129 participants who answered all insomnia symptom related questions in these analyses.

### Study design

A detailed description of this study's methods has been previously reported [21]. Most relevant to these analyses, eligible participants completed a REDCAP-administered survey either remotely or while in the clinic for routine visits. Part 1 of the survey assessed demographic information and substance use history. Part 2 asked participants to complete a shortened version of NIDA's Phenotyping Assessment Battery (i.e., PhAB-Brief) inclusive of neurofunctional assessments targeting domains inherent to addiction [21]. Participants were compensated $15. Participants provided written informed consent and study procedures were approved by the IRB at Virginia Commonwealth University (HM20023390).

### Primary outcome: Insomnia

The Insomnia Severity Index (ISI) [22] is a 7-item instrument that measures patient-reported severity of insomnia symptoms and has been used extensively to track insomnia treatment progress in clinical trials [8, 22, 23]. Raw scores (range: 0–28) on the ISI were dichotomized to create two groups based on prior work that determined an optimal cut-off point for identifying diagnosable insomnia in clinical samples, one with minimal evidence of clinically significant insomnia symptoms (INS-; ISI $\leq$ 10) and a second with evidence of clinically significant insomnia symptoms (INS+; ISI $\geq$ 11) [23]. Others have suggested using a higher cut-off point for determining insomnia status ($\geq$15; "moderate-to-severe" insomnia symptoms) [24], which we considered in sensitivity analyses. We also considered the sensitivity of our estimates to modeling ISI scores as a continuous rather than categorical variable. However, the categorical definition with the lower cut-off point was adopted as the preferred specification in subsequent analyses because one of our goals in these analyses was to determine the prevalence of clinically relevant insomnia symptoms within our sample regardless of severity.

### Neurofunctional measures

The PhAB-Brief assesses six neurofunctional domains deemed critical to addiction: sleep (Insomnia Severity Index [ISI] [22]), negative emotionality (Distress Tolerance Scale [DTS] [25], Patient Health Questionnaire-9 [PHQ-9] [26], General Anxiety Disorder-7 [GAD-7]

[27], PROMIS-10 Global Mental Health Scale [28], Post-Traumatic Stress Disorder Checklist [PCL-5] [29], Buss-Perry Aggression Scale [BPAS] [30], Snaith-Hamilton Pleasure Scale [SHPS] [31]), metacognition (Metacognition Questionnaire-30 [MCQ-30] [32]), interoception (Multidimensional Assessment of Interoceptive Awareness [MAIA] [33]), cognition (5-Trial Adjusting Delay Discounting Task [5-DD] [34]), and reward (Short Impulsive Behavior Scale [SUPPS-P] [35]). We describe each of these individual measures in detail, as well as the development of the PhAB-Brief, elsewhere [21]. In general, we adhered to the standard reporting practices associated with each measure. One exception to this practice is that the key measure derived from the 5-DD was the Effective-Delay-50 (ED50; time required for a reward to diminish in subjective value by 50%), which was generated by taking the reciprocal of the observed discounting rate (S1 Appendix).

## Other variables of interest

Participants reported their gender, age, race, ethnicity, educational attainment, employment status, buprenorphine dose, and start date of buprenorphine therapy. Finally, participants responded on a Likert-like scale (0 [not at all]– 4 [very much]) to indicate whether they felt sleep interfered with their OUD treatment and whether improved sleep would assist in their OUD treatment.

## Data preparation and analyses

Chi-square tests assessed differences in categorical outcomes between the two insomnia symptom groups. For continuous variables, because many of the neurofunctional measures followed a non-normal distribution (per Shapiro-Wilk tests), non-parametric Mann-Whitney U-tests compared each neurofunctional measure between the two insomnia symptom groups. To adjust for possible confounding of the significant bivariate relationships observed, linear regressions modeled each neurofunctional key variable as a function of insomnia status controlling for respondent's gender, age, race, education, and current buprenorphine dose. Additional sensitivity analyses utilized multivariate normal regression to impute missing values. All statistical analyses were performed in Stata SE 17 (College Station, TX) and alpha was set to 0.05. Because this study was conducted to inform hypotheses for future clinical trials, all statistical comparisons were pre-planned, and we did not seek to test "omnibus" hypotheses, we report exact p-values from each test instead of directly correcting for multiple comparisons [36]. Though no formal power analyses were done *a priori*, we exceeded our initial planned sample size of 100 individuals (50 women, 50 men) due to accruing women at a faster rate than men.

## Results

### Sample description

Overall, respondents (N = 129) identified predominately as cisgender women (62%), white (55.8%), unemployed (44.2%), and on a sublingual buprenorphine dose of $\geq 16$ mg/day (47.3%). Roughly 60% of participants reported clinically significant insomnia symptoms. Due to the group definitions adopted here, the mean (standard deviation [SD], range) ISI score for the "minimal evidence of insomnia symptoms" group (INS-) was 5.26 (3.3, 0–10) and for the "evidence of insomnia symptoms" group (INS+) was 18.72 (4.6, 11–28). On average (SD, range), participants in this sample had started buprenorphine therapy 2.52 (2.30, 0.12–13.20) years before the time of data collection. Demographic characteristics did not significantly differ between the two insomnia symptom status groups (Table 1).

**Table 1. Sociodemographic and key clinical variables by insomnia symptom status group.**

| | Overall | Minimal Evidence of Insomnia Symptoms (INS-) (ISI: ≤10) | Evidence of Insomnia Symptoms (INS+) (ISI: ≥11) | p |
|---|---|---|---|---|
| **N (%)** | 129 (100%) | 53 (41.1%) | 76 (58.9%) | - - - |
| **Gender** | | | | 0.130 |
| *Cisgender women* | 79 (61.7%) | 28 (53.9%) | 51 (67.1%) | |
| *Cisgender men* | 50 (38.3%) | 25 (46.2%) | 25 (32.9%) | |
| **Age** (years) | 40.18 (9.9) [N = 115] | 40.18 (9.9) [N = 47] | 39.51 (9.4) [N = 68] | 0.418 |
| **Race** | | | | 0.666 |
| *American Indian* | 3 (2.3%) | 1 (1.9%) | 2 (2.6%) | |
| *Black or African American* | 46 (35.6%) | 23 (43.4%) | 23 (30.3%) | |
| *Native Hawaiian or Pacific Islander* | 1 (0.8%) | 0 (0.0%) | 1 (1.3%) | |
| *White* | 72 (55.8%) | 26 (49.1%) | 46 (60.5%) | |
| *More than one race* | 4 (3.1%) | 2 (3.8%) | 2 (2.6%) | |
| *Other* | 3 (2.3%) | 1 (1.9%) | 2 (2.6%) | |
| **Ethnicity** (N = 108) | | | | 0.068 |
| *Hispanic or Latinx* | 5 (4.6%) | 0 (0.0%) | 5 (7.6%) | |
| *Not Hispanic or Latinx* | 103 (95.4%) | 42 (100%) | 61 (92.4%) | |
| **Employment** | | | | 0.610 |
| *Employed* | 47 (36.4%) | 17 (32.1%) | 30 (39.5%) | |
| *Unemployed* | 57 (44.2%) | 24 (45.3%) | 33 (43.4%) | |
| *Disabled* | 25 (19.4%) | 12 (22.6%) | 13 (17.1%) | |
| **Education** | | | | 0.057 |
| *Did not finish high school* | 26 (20.2%) | 16 (30.2%) | 10 (13.2%) | |
| *High school diploma or GED* | 59 (45.7%) | 22 (41.5%) | 37 (48.7%) | |
| *At least some postsecondary education* | 44 (34.1%) | 15 (28.3%) | 29 (38.2%) | |
| **Buprenorphine Dose** | | | | 0.134 |
| *≤8 mg (sublingual)* | 14 (10.9%) | 8 (15.1%) | 6 (7.9%) | |
| *9–16 mg (sublingual)* | 40 (31.0%) | 17 (32.1%) | 23 (30.3%) | |
| *≥16 mg (sublingual)* | 61 (47.3%) | 21 (39.6%) | 40 (52.6%) | |
| *100 mg (extended-release injection)* | 11 (8.5%) | 7 (13.2%) | 4 (5.3%) | |
| *300 mg (extended-release injection)* | 3 (2.3%) | 0 (0.0%) | 3 (4.0%) | |
| ***Time since starting buprenorphine (years)*** | 2.52 (2.3) | 2.55 (2.2) | 2.50 (2.4) | 0.727 |

NOTE: For continuous variables, mean and standard deviation (SD) [N] are presented and insomnia severity index (ISI) groups were compared using u-tests. For categorical variables, subsample size (n) and percentage of column are presented; ISI groups were compared using chi-squared tests.

## Neurofunctional correlates of insomnia symptoms

For neurofunctional measures (Table 2), within the negative emotionality domain, INS+ participants were more likely to endorse symptoms of poorer overall mental health, depression, anxiety, and post-traumatic stress than INS- participants (ps<0.001). Moreover, the INS+ participants were more intolerant to stress, exhibited lower acceptance of stress and trust in stress-coping skills, as well as reported being more absorbed by distressing emotions (ps<0.01). In the Metacognition and Interoception domains, evidence of clinically significant insomnia was associated with having negative beliefs about uncontrollable and dangerous thoughts (p<0.05) as well as heightened awareness of bodily sensations (p<0.01). No statistically significant differences were observed within the reward and cognition domains between

**Table 2. Unadjusted differences in key neurofunctional measures as a function of insomnia status group.**

| Measure | Overall | Minimal Evidence of Insomnia Symptoms (INS-) (ISI: ≤10) | Evidence of Insomnia Symptoms (INS+) (ISI: ≥11) | p |
|---|---|---|---|---|
| **Neurofunctional Domain 1: Reward** | | | | |
| **Short UPPS-P Impulsive Behavior Scale** | | | | |
| *Negative urgency* | 10.08 (3.2) [N = 117] | 9.46 (2.9) [N = 49] | 10.51 (3.4) [N = 68] | 0.112 |
| *Lack of perseverance* | 7.05 (2.3) [N = 120] | 6.71 (1.8) [N = 49] | 7.28 (2.5) [N = 71] | 0.377 |
| *Lack of premeditation* | 6.86 (2.4) [N = 117] | 6.64 (2.2) [N = 50] | 7.03 (2.6) [N = 67] | 0.537 |
| *Sensation seeking* | 8.46 (3.3) [N = 113] | 8.00 (2.8) [N = 45] | 8.78 (3.5) [N = 68] | 0.402 |
| *Positive urgency* | 9.64 (3.2) [N = 118] | 9.31 (2.6) [N = 49] | 9.88 (3.4) [N = 69] | 0.305 |
| **Neurofunctional Domain 2: Cognition** | | | | |
| **5-Trial Adjusting Delay Discounting Task (Effective Delay 50)** | 1822.98 (2497.0) [N = 120] | 1715.78 (2628.3) [N = 50] | 1899.55 (2415.2)) [N = 70] | 0.693 |
| **Neurofunctional Domain 3: Interoception** | | | | |
| **Multidimensional Assessment of Interoceptive Awareness** | | | | |
| *Noticing* | 3.26 (1.3) [N = 119] | 2.86 (1.4) [N = 49] | 3.55 (1.2) [N = 70] | 0.010 |
| *Not distracting* | 2.47 (1.1) [N = 120] | 2.68 (1.1) [N = 50] | 2.31 (1.1) [N = 70] | 0.137 |
| *Not worrying* | 2.50 (0.9) [N = 120] | 2.57 (1.0) [N = 49] | 2.45 (0.9) [N = 71] | 0.299 |
| *Attention regulation* | 2.65 (1.2) [N = 118] | 2.64 (1.3) [N = 47] | 2.65 (1.2) [N = 71] | 0.955 |
| *Emotional awareness* | 3.23 (1.3) [N = 117] | 2.97 (1.5) [N = 50] | 3.43 (1.0) [N = 67] | 0.138 |
| *Self-regulation* | 2.45 (1.3) [N = 117] | 2.38 (1.4) [N = 47] | 2.50 (1.2) [N = 70] | 0.660 |
| *Body listening* | 2.31 (1.2) [N = 118] | 2.16 (1.3) [N = 49] | 2.42 (1.1) [N = 69] | 0.239 |
| *Trusting* | 2.97 (1.4) [N = 118] | 3.14 (1.6) [N = 49] | 2.85 (1.3) [N = 69] | 0.222 |
| **Neurofunctional Domain 4: Negative Emotionality** | | | | |
| **Patient Health Questionnaire-9** | 10.37 (6.6) [N = 117] | 6.7 (4.9) [N = 48] | 12.94 (6.5) [N = 69] | <0.001 |
| **General Anxiety Disorder-7** | 8.37 (6.2) [N = 119] | 5.54 (5.1) [N = 48] | 10.28 (6.2) [N = 71] | <0.001 |
| **Post-Traumatic Stress Disorder Checklist** | 33.30 (21.3) [N = 111] | 24.57 (19.0) [N = 47] | 39.87 (20.7) [N = 64] | <0.001 |
| **PROMIS-10 Health Scale** | | | | |
| *Global physical health* | 11.46 (1.9) [N = 126] | 11.82 (1.9) [N = 51] | 11.21 (1.9) [N = 75] | 0.161 |
| *Global mental health* | 12.15 (2.4) [N = 126] | 11.53 (2.3) [N = 51] | 12.57 (2.5) [N = 75] | 0.018 |
| **Distress Tolerance Scale** | | | | |
| *Tolerance* | 3.02 (1.1) [N = 119] | 2.71 (1.1) [N = 48] | 3.23 (1.0) [N = 71] | 0.009 |
| *Absorbance* | 2.89 (1.1) [N = 117] | 2.56 (1.2) [N = 48] | 3.12 (1.1) [N = 69] | 0.006 |
| *Appraise* | 2.78 (0.8) [N = 113] | 2.56 (0.9) [N = 49] | 2.95 (0.8) [N = 64] | 0.009 |
| *Regulation* | 2.87 (1.0) [N = 117] | 2.81 (1.1) [N = 49] | 2.92 (1.0) [N = 68] | 0.570 |
| **Buss-Perry Aggression Scale** | 71.93 (21.0) [N = 102] | 74.61 (17.6) [N = 41] | 70.13 (23.0) [N = 61] | 0.400 |
| *Physical aggression* | 22.95 (7.3) [N = 112] | 24.09 (6.2) [N = 47] | 22.12 (7.9) [N = 65] | 0.185 |
| *Verbal aggression* | 12.93 (4.3) [N = 118] | 13.16 (3.5) [N = 49] | 12.77 (4.9) [N = 69] | 0.624 |
| *Anger* | 16.58 (5.7) [N = 116] | 17.23 (5.3) [N = 48] | 16.12 (5.9) [N = 68] | 0.192 |
| *Hostility* | 20.22 (8.5) [N = 113] | 19.76 (7.1) [N = 46] | 20.54 (9.3) [N = 67] | 0.717 |
| **Snaith-Hamilton Pleasure Scale** | 2.29 (2.7) [N = 129] | 2.15 (2.9) [N = 53] | 2.39 (2.5) [N = 76] | 0.291 |

*(Continued)*

**Table 2.** (Continued)

| Measure | Overall | Minimal Evidence of Insomnia Symptoms (INS-) (ISI: ≤10) | Evidence of Insomnia Symptoms (INS+) (ISI: ≥11) | p |
|---|---|---|---|---|
| **Neurofunctional Domain 5: Metacognition** | | | | |
| **Metacognition Questionnaire-30** | 65.43 (16.0) [N = 108] | 62.27 (15.8) [N = 45] | 67.68 (2.0) [N = 63] | 0.135 |
| *Lack of confidence* | 11.97 (4.2) [N = 117] | 11.29 (3.8) [N = 48] | 12.43 (4.5) [N = 69] | 0.237 |
| *Positive worry* | 10.66 (4.1) [N = 117] | 10.15 (3.8) [N = 48] | 11.01 (4.3) [N = 69] | 0.367 |
| *Cognitive confidence* | 15.91 (4.3) [N = 117] | 15.38 (4.8) [N = 47] | 16.26 (4.0) [N = 70] | 0.361 |
| *Uncontrollability and danger* | 13.74 (4.8) [N = 117] | 14.59 (4.7) [N = 48] | 14.59 (4.7) [N = 69] | 0.035 |
| *Controlling thoughts* | 13.38 (4.3) [N = 116] | 13.16 (4.1) [N = 49] | 13.54 (4.4) [N = 67] | 0.820 |

NOTE: A total of 129 participants were included in these analyses, though the sample size for each outcome varies due to participants not answering every question for a given instrument (i.e., list-wise deletion). For continuous variables, raw mean and standard deviation (SD) [N] are presented and ISI groups were compared using U-tests. Group differences in adjusted mean scores after controlling for demographic variables and buprenorphine dose are presented in S3 Table in S1 Appendix. The Distress Tolerance Scale was reverse coded, such that higher scores indicate more problematic stress reactions within each subdomain. Bolded lines indicate total or summary scores for scales utilized in the PhAB-Brief; italicized lines indicate scores from subscales.

INS groups (ps>0.05). Results were insensitive to imputing missing values (S2 Table in S1 Appendix), controlling for possible demographic confounders (S3 Table in S1 Appendix), adopting a higher ISI cutoff score (S4 Table in S1 Appendix), or modeling ISI scores as a continuous variable in covariate-adjusted regressions (S5 Table in S1 Appendix).

## Perspectives on the influence of sleep on OUD treatment & recovery

The INS+ group more strongly agreed with the statements that "poor sleep interferes with OUD treatment" (1.41 [1.3] vs. 0.50 [0.9]) and that "improved sleep would assist with OUD treatment" (2.53 [1.3] vs. 1.37 [1.4]) than the INS- group (ps<0.001).

## Discussion

Our findings indicate that among a clinical sample of individuals with OUD receiving buprenorphine, clinically meaningful insomnia symptoms are a prevalent issue and one identified by patients as a barrier to OUD treatment progress. Moreover, patients who experienced a higher burden of insomnia symptoms also reported significantly higher levels of depression, anxiety, post-traumatic stress, stress intolerance, unhelpful metacognition, and interoceptive awareness. Participants experiencing a higher burden of insomnia symptoms in this sample also showed potentially meaningful increases in aspects of impulsivity, such as "negative urgency" (i.e., tendency to act rashly when distressed), though between group differences in these measures did not achieve statistical significance. Our results highlight how insomnia symptoms can persist in patients with OUD receiving buprenorphine for years, especially when not addressed directly.

Because the neurofunctional correlates of insomnia symptom severity identified here are also associated with addiction treatment outcomes [9, 10, 37], we speculate that they represent a mechanism through which poor sleep contributes to an increased risk of return to substance use in OUD patients. Thus, although correlative, these data indicate that combining MOUD

treatment with strategies that address insomnia symptoms might help to improve treatment outcomes. For example, pharmacological (e.g., orexin receptor antagonists) or behavioral (e.g., cognitive behavioral therapy for insomnia) interventions could be leveraged in the design of personalized treatments for patients with OUD alongside medications for OUD (i.e., buprenorphine). Given the diverse ways in which sleep disturbances can arise and impact substance use behaviors, developing personalized treatment strategies in this OUD populations could be a powerful approach to improving outcomes by considering the unique barriers to recovery that individual patients may face. Studies testing this possibility would also shed light on the causal relationship (and directionality) between sleep dysregulation and OUD treatment outcomes.

Our results are largely consistent with previous literature linking poor sleep quality to negative affect, stress reactivity, and pain sensitivity, but expand knowledge by drawing upon a diverse outpatient population currently receiving MOUD and utilizing a multidimensional assessment of neurofunction. Our results also synergize with previous literature describing elevated autonomic responses to relived distressing experiences among people with insomnia, suggesting that increased experiences of negative emotionality and struggles with managing the resulting stress are potentially important consequences of insomnia in OUD treatment populations [38]. Deficits in neurofunction identified using emotional data are a potentially distinct consequence of insomnia [39] from deficits in other areas of neurofunction such as procedural learning, attentiveness, and executive functioning [40]. The insomnia-emotionality link may be especially salient in substance use populations, which tend to have high rates of previous trauma and comorbid psychiatric conditions [41, 42]. Additionally, the neurofunctional differences by insomnia symptom status identified in this study largely align with the domains represented in the predominant neurofunctional phenotype our group recently [43] identified among a large sample of individuals with addiction using exploratory factor analysis of NIDA PhAB data (i.e., sleep, negative emotionality, metacognition, reward). Although not studied here, it is interesting to note that the orexin/hypocretin system is a common regulator of each of these functional outcomes [44, 45] and may therefore serve as a common biological mediator of the effects observed here. Indeed, orexin neurons are upregulated in response to morphine and heroin [46] which is expected to have important implications for sleep, stress and cognitive outcomes [47]. Overall, these findings highlight how sleep disturbances may be part of a constellation of symptoms with substantial neurofunctional overlap in patients with OUD, possibly via neurobiological interconnections such as the orexin system—future work is required to test this hypothesis [13, 14].

These results should be considered in light of several limitations. The use of a cross-sectional survey meant that we were unable to obtain specific evidence that the sleep disturbances in the INS+ group played a causal role in the emergence of other symptomatology; future studies should seek to test if changes in insomnia symptoms (including improvements associated with pharmacological or psychological interventions) are related to changes in neurofunctional phenotype and OUD treatment outcomes over time. Investigations of the patterns of comorbidity of mental illnesses [48] and their shared genetic [49] and neurodevelopmental [50, 51] architecture point to the existence of a common "p-factor" that nonspecifically indexes overall mental health burden in an individual, where sleep disturbance is thought to be a component of the p-factor [52]. By extension, poor sleep in the more symptomatic MOUD patients could simply be a manifestation of this; in this case, sleep interventions may be expected to have less impact compared to a scenario where sleep directly modulates mental health/addiction outcomes. In addition, due to exclusion of patients with severe comorbid mental illness, our findings may not generalize to other treatment contexts, populations, or sleep disorders. However, it is likely that inclusion of MOUD patients with more psychiatric comorbidity (i.e.

higher p-factor) could have yielded even stronger results. The null results in the "cognition" domain of the PhAB-Brief documented here may be attributable to our use of a *temporal* discounting task instead of an *effort-based* discounting task [16]; future work should explore differences in discounting task designs and seek to understand how insomnia may differentially impact each of these impulsivity measures. Participant responses on some items may have been impacted by recall and social desirability biases [53] and the relatively small sample size limits the generalizability of our findings. Finally, these analyses did not correct for family-wise errors which raises the possibility of having reported some false-positive results. Results reported here should thus be considered as "hypothesis-generating" rather than confirmatory. Despite these limitations, our findings may help explain why patients with sleep disturbances experience worse OUD treatment outcomes than their counterparts. Our data also highlight the need for future work to identify biological systems that might mediate sleep dysregulation during OUD treatment, which might serve as targets for novel treatments to improve treatment outcomes.

## Conclusion

We report that the subset of MOUD patients with sleep problems (i.e., insomnia) in the clinically-significant range show worse negative affect symptomatology, poorer frustration tolerance, and other features not conducive to OUD recovery. This suggests that the link between impaired sleep and psychiatric illness severity found in other clinical contexts and populations is also evident in the MOUD population. Improved understanding of the linkages between sleep disturbances, neurofunction, and substance use behaviors may be useful in the development of novel, individualized treatment strategies for individuals with OUD and comorbid insomnia. Future studies could expand upon these findings by measuring changes in sleep, neurofunction, and substance use behaviors over time (such as after interventions that target sleep), as well as by deploying neuroimaging and other modalities to undercover mechanisms of shared symptomatology between sleep and collateral problems in SUD populations. Finally, our data suggests that insomnia could be a symptom target in future clinical trials in patients with opioid use disorder in treatment with buprenorphine.

## Supporting information

**S1 Appendix. Supporting information.** Contains all supporting information (S1–S5 Tables). (DOCX)

## Acknowledgments

We would like to thank the participants of this study for sharing their time and insights with our team.

## Author Contributions

**Conceptualization:** Augustus M. White, Michelle Eglovitch, Anna Beth Parlier-Ahmad, Joseph M. Dzierzewski, Morgan James, James M. Bjork, F. Gerard Moeller, Caitlin E. Martin.

**Data curation:** Michelle Eglovitch, Anna Beth Parlier-Ahmad, Caitlin E. Martin.

**Formal analysis:** Augustus M. White, Michelle Eglovitch, James M. Bjork.

**Funding acquisition:** F. Gerard Moeller, Caitlin E. Martin.

**Investigation:** Joseph M. Dzierzewski, Caitlin E. Martin.

**Methodology:** Augustus M. White, Michelle Eglovitch, Anna Beth Parlier-Ahmad, Joseph M. Dzierzewski, Morgan James, James M. Bjork, F. Gerard Moeller, Caitlin E. Martin.

**Project administration:** Michelle Eglovitch, F. Gerard Moeller, Caitlin E. Martin.

**Resources:** F. Gerard Moeller, Caitlin E. Martin.

**Software:** Augustus M. White, F. Gerard Moeller, Caitlin E. Martin.

**Supervision:** F. Gerard Moeller, Caitlin E. Martin.

**Validation:** Caitlin E. Martin.

**Visualization:** Augustus M. White.

**Writing – original draft:** Augustus M. White, Michelle Eglovitch, Anna Beth Parlier-Ahmad, Caitlin E. Martin.

**Writing – review & editing:** Augustus M. White, Michelle Eglovitch, Anna Beth Parlier-Ahmad, Joseph M. Dzierzewski, Morgan James, James M. Bjork, F. Gerard Moeller, Caitlin E. Martin.

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
