## [Decision Letter · Decision Letter 0]

18 Mar 2024

PONE-D-23-39561Insomnia symptoms and neurofunctional correlates among adults receiving buprenorphine for opioid use disorderPLOS ONE

Dear Dr. White,

Thank you for submitting your manuscript to PLOS ONE. After careful consideration, we feel that it has merit but does not fully meet PLOS ONE’s publication criteria as it currently stands. Therefore, we invite you to submit a revised version of the manuscript that addresses the points raised during the review process.

We look forward to receiving your revised manuscript.

Kind regards,

Sungwoo Lim, DrPH

Academic Editor

PLOS ONE

Reviewers' comments:

Reviewer's Responses to Questions

**Comments to the Author**

1. Is the manuscript technically sound, and do the data support the conclusions?

Reviewer #1: Yes

2. Has the statistical analysis been performed appropriately and rigorously? 

Reviewer #1: Yes

3. Have the authors made all data underlying the findings in their manuscript fully available?

Reviewer #1: Yes

4. Is the manuscript presented in an intelligible fashion and written in standard English?

Reviewer #1: Yes

5. Review Comments to the Author

Reviewer #1: See attachment ........................................................................................................................................... ................................................................................................................................................................

6. PLOS authors have the option to publish the peer review history of their article (what does this mean?). If published, this will include your full peer review and any attached files.

Reviewer #1: **Yes: **Allison K. Wilkerson

---

## [Author Response · Author response to Decision Letter 0]

2 May 2024

Please see attached documentation for our full response to reviewers. For convenience, we've copy-and-pasted our responses below.

-

Insomnia Symptoms and Neurofunctional Correlates Among Adults Receiving Buprenorphine for Opioid Use Disorder

Reviewer Response Document

PLOS ONE – Personalized Medicine Special Collection (Manuscript ID: PONE-D-23-39561)

26 March 2024

To Editorial Staff of PLOS ONE:

We would like to thank the reviewer and editor for their consideration of our manuscript and for extending the opportunity to make revisions and resubmit. We believe that the revisions detailed in this response document and in the updated manuscript/supplementary materials improve the rigor and relevance of our work. We have aimed to be as responsive as possible to critiques that emerged during the peer-review process (see below), but if any further information is needed before making a final determination on the editorial status of this manuscript please do not hesitate to contact us. 

Additionally, we’d like to note that during the revision process we identified a small coding error that affected the results from the 5-trial adjusting delay discounting task. The coding error has now been corrected and all affected tables have been updated in the revised manuscript. The impact of the coding error on our estimates was relatively minor in magnitude and did not change the overall results/conclusions presented in our manuscript.

Thank you again for your consideration of our manuscript for publication in PLOS ONE’s special collection on Personalized Medicine and we look forward to hearing from you regarding next steps. 

Sincerely, 

Augustus M. White, BA

Michelle Eglovitch, MPH MS

Anna Beth Parlier-Ahmad, MS

Joseph M Dzierzewski, PhD

Morgan James, PhD

James M. Bjork, PhD

F. Gerard Moeller, MD

Caitlin E. Martin, MD-MPH

 

Reviewer: 1

The authors of this manuscript aimed to examine the relationship between insomnia symptoms and several areas of neurocognitive performance and mental health symptoms in persons with opioid use disorder (OUD) who are receiving buprenorphine. The authors review the growing body of literature on various aspects of sleep disruption, including insomnia, in those who are receiving medications for opioid use disorder and provide rationale for the need to examine how this disruption relates to neurofunctional correlates. A continuous scale of insomnia was dichotomized by a cutoff score to differentiate those with and without clinically significant insomnia symptoms. This could be seen as a limiting factor, though the authors provided rationale and pointed to precedent in other literature. In 129 persons receiving buprenorphine they found the majority (58.9%) were experiencing clinical significant insomnia symptoms, and higher symptoms were related to symptoms of depression, anxiety, post-traumatic stress, stress intolerance, unhelpful metacognition, and interoceptive awareness. Further, those with significant insomnia symptoms endorsed sleep as interfering with OUD treatment and that improving sleep would assist with OUD treatment. Strengths of the manuscript include the use of validated measures, a fairly large sample size, and addressing the understudied area of the relationship between sleep and neurofunctioning in OUD recovery. Moreover, the authors identified a specific sleep problem (insomnia) rather than generically discussing “sleep disturbance” or “poor sleep.” This nuance is very important in the literature, as the more broad terms do little to advance the field. Clarifying which sleep disorder symptoms are being assessed helps focus to be narrowed to specific, targetable modifiable risk factors that can be addressed in future studies. 

We would like to thank the reviewer for their careful consideration of our submission and agree that the measure set, sample, and focus on a specific sleep problem featured in this study are among the manuscript’s strengths. Furthermore, we would like to thank the reviewer for the constructive critiques of the manuscript they offered. Our responses to each of the points raised during the peer-review process are detailed below and we feel that our revisions have improved the manuscript considerably. 

We also agree with the reviewer that use of a dichotomized insomnia symptom severity measure is a potential limitation of our approach. To assess the sensitivity of our conclusions to the functional form of the insomnia classification variable, in our original submission we did model insomnia symptoms as a continuous predictor of neurofunctional outcomes (Supplementary Table 5). Sensitivity analyses suggested that the results presented in our original submission were similar under the dichotomized and continuous specifications of insomnia symptom severity. Due to the similarity in results obtained under the dichotomized and continuous specifications of the insomnia symptom severity variable and the precedent set in previous literature for using categorical specifications of insomnia symptom severity, we have retained the dichotomized functional form as the primary specification in this revised manuscript. 

This manuscript could be improved if the authors considered and/or edited as follows:

 If word count allows, this paper would benefit from a little more explanation in the Introduction. Specifically P. 3, paragraph 2, lines 55-58 references that neurofunctional performance can be impacted and minimal research has been done in this area, but does not break down the various aspects of functioning/performance that will be looked at and why. This is the gap in the literature this study is proposing to fill and should therefore probably be expanded a little more. 

We welcome the opportunity to further expand upon the gaps in the literature that our study attempts to address and agree that this would improve the readability of the manuscript. Being mindful of the intended word count for this manuscript, we have amended our introduction as follows (changes in bold):

Previous evidence indicates that poor sleep contributes to the development and maintenance of substance use disorders by heightening pain sensitivity, promoting negative affect, and adversely impacting stress reactivity as well as self-care-related executive functions.[1, 4, 9-11] These relationships may arise via a depletion of “cognitive resources”[12] or disruption of neurobiological systems important for arousal, such as imbalance in the orexin/hypocretin system.[3, 13, 14] In humans, these disruptions can manifest as differences in incentive-related decision-making that can be measured using validated metrics.[9, 13, 15-17] Moreover, sleep quality can impact how individuals choose to respond (e.g., to avoid) exertion of effort [16], which could be a mechanism for poorer performance on a variety of tasks. Sleep quality may thus influence a set of neurofunctional domains that are believed to predict substance use treatment outcomes.[21] However, previous studies have typically considered only a narrow set of neurofunctional indices within the context of a single design or relied on samples whose subjects were actively engaged in substance use. Thus, the relationship between sleep and broad neurofunctional outcomes in addiction treatment samples is not well characterized. 

Here, we attempt to bolster the existing literature by examining the prevalence of insomnia and its neurofunctional correlates in the domains of reward, cognition, interoception, negative emotionality, and metacognition among a sample of individuals receiving buprenorphine for OUD. Sleep disturbances may alter the underlying neurofunctional mechanisms of OUD at various stages of OUD treatment and recovery. Obtaining insights on how sleep disturbance relates to other symptomatology would be useful in the creation of individualized treatment plans for patients with OUD and comorbid insomnia by identifying potential targets for intervention[9] such as adjunctive pharmacologic (e.g., orexin receptor antagonists, antidepressants, etc.) and non-pharmacologic (e.g., CBT-I, working memory training, cue exposure therapy, etc.) treatments to medications for OUD. Given the multi-factorial (e.g., genetic, environmental, lifestyle) etiology of both sleep disturbances and substance use behaviors,[18] the capacity for personalization in treatment for this population represents a promising avenue to improve OUD outcomes.[1, 19, 20] (page 3)

 On page 5 the reporting of ISI “INS+” and “INS-“ randomly switches to serif font rather than sans serif

Thank you for calling out this oversight in the typesetting of our original submission. In the revised manuscript we have ensured that Arial font is used throughout (changes in bold). 

“Raw scores (range: 0-28) on the ISI were dichotomized to create two groups based on prior work that determined an optimal cut-off point for identifying diagnosable insomnia in clinical samples, one with minimal evidence of clinically significant insomnia symptoms (INS-; ISI ≤ 10) and a second with evidence of clinically significant insomnia symptoms (INS+; ISI ≥ 11).” (Page 5)

 It has been proposed by some that sleep should stabilize over time after dose is stabilized, thus how long they have been receiving buprenorphine would impact sleep, and in theory these other variables would improve. If the authors have data on how long participants have been engaged with MOUD it may be helpful to present the mean amount of time or range of time. That would allow demonstration that insomnia can persist for years if not directly addressed.

We agree that the stage of OUD treatment with buprenorphine is a potentially important variable to consider in this manuscript. The inclusion criteria for this study required all participants to have received buprenorphine for at least 6 weeks before enrolling. Thus, we consider participants in our sample to have been on a “stabilized” dose of buprenorphine at the time of data collection. 

While we did not collect data that could speak to how long participants had been maintained on their current dose of buprenorphine, we did ask participants to report when they first began receiving buprenorphine. By taking the difference in the time since starting buprenorphine and consenting to participate, we report that participants in our sample had been on buprenorphine for around 2.5 years on average in both groups (range: 0.117 years to 13.2 years). We have amended the text of the results section as well as Table 1 in our manuscript to reflect this information (see below [changes in bold]). 

“Due to the group definitions adopted here, the mean (standard deviation, range) ISI score for the “minimal evidence of insomnia symptoms” group (INS-) was 5.26 (3.3, 0-10) and for the “evidence of insomnia symptoms” group (INS+) was 18.72 (4.6, 11-28). On average (SD, range), participants in this sample had started buprenorphine therapy 2.52 (2.30, 0.12-13.20) years before the time of data collection. Demographic characteristics did not significantly differ between the two insomnia symptom status groups (Table 1)”. (page 7)

TABLE 1. Sociodemographic and Key Clinical Variables by Insomnia Symptom Status Group (abbreviated) 

 Overall

 Minimal Evidence of Insomnia Symptoms 

(INS-)

(ISI: ≤10) Evidence of Insomnia Symptoms 

(INS+)

(ISI: ≥11) p

N (%) 129 (100%) 53 (41.1%) 76 (58.9%) ---

Time since starting buprenorphine (years) 2.52 (2.30) 2.55 (2.21) 2.50 (2.43) 0.727

NOTE: For continuous variables, mean and standard deviation (SD) [N] are presented and insomnia severity index (ISI) groups were compared using u-tests. For categorical variables, subsample size (n) and percentage of column are presented; ISI groups were compared using chi-squared tests. 

We have also revised the following sentence in the discussion section to highlight the point raised by the reviewer about the persistence of insomnia symptoms in this population (changes in bold):

“Our findings indicate that among a clinical sample of individuals with OUD receiving buprenorphine, clinically meaningful insomnia symptoms are a prevalent issue and one identified by patients as a barrier to OUD treatment progress. Moreover, patients who experienced a higher burden of insomnia symptoms also reported significantly higher levels of depression, anxiety, post-traumatic stress, stress intolerance, unhelpful metacognition, and interoceptive awareness. Participants experiencing a higher burden of insomnia symptoms in this sample also showed potentially meaningful increases in aspects of impulsivity, such as “negative urgency” (i.e., tendency to act rashly when distressed), though between group differences in these measures did not achieve statistical significance. Our results highlight how insomnia symptoms can persist in patients with OUD receiving buprenorphine for years, especially when not addressed directly” (Page 10). 

 Similar to the Introduction, the Discussion would benefit from elaboration. The non-significant findings are just as interesting as the significant ones. There is a good bit of literature about how insomnia often correlates with subjective measures of cognitive performance and mental health symptoms, but the relationship with objective measures is less clear. That seems to have been mirrored in this group as well and hypotheses around that and its implications for treatment would enrich the Discussion section. 

We would like to thank the reviewer for challenging us to think more deeply about our results and to better contextualize our findings with the extant literature. We have attempted to expand on our original discussion of the study results may calling specific attention to intriguing but non-significant results, contrasting deficits in neurofunction that rely on emotional data with other measures of neurofunction, and discussing differences in the tasks used in this study with previous work. Below we present passages that reflect these amendments (changes in bold). 

“Our findings indicate that among a clinical sample of individuals with OUD receiving buprenorphine, clinically meaningful insomnia symptoms are a prevalent issue and one identified by patients as a barrier to OUD treatment progress. Moreover, patients who experienced a higher burden of insomnia symptoms also reported significantly higher levels of depression, anxiety, post-traumatic stress, stress intolerance, unhelpful metacognition, and interoceptive awareness. Participants experiencing a higher burden of insomnia symptoms in this sample also showed potentially meaningful increases in aspects of impulsivity, such as “negative urgency” (i.e., tendency to act rashly when distressed), though between group differences in these measures did not achieve statistical significance. Our results highlight how insomnia symptoms can persist in patients with OUD receiving buprenorphine for years, especially when not addressed directly” (page 10)

“Our results are largely consistent with previous literature linking poor sleep quality to negative affect, stress reactivity, and pain sensitivity, but expand knowledge by drawing upon a diverse outpatient population currently receiving MOUD and utilizing a multidimensional assessment of neurofunction. Our results also synergize with previous literature describing elevated autonomic responses to relived distressing experiences among people with insomnia, suggesting that increased experiences of negative emotionality and struggles with managing the resulting stress are potentially important consequences of insomnia in OUD treatment populations.[39] Deficits in neurofunction identified using emotional data are a potentially distinct consequence of insomnia[40] from deficits in other areas of neurofunction such as procedural learning, attentiveness, and executive functioning [41]. The insomnia-emotionality link may be especially salient in substance use populations which tend to have high rates of previous trauma and comorbid psychiatric conditions [42, 43]. Additionally, the neurofunctional differences by insomnia symptom status identified in this study largely align with the domains represented in the predominant neurofunctional phenotype our group recently[44] identified among a large sampl

---

## [Decision Letter · Decision Letter 1]

14 May 2024

Insomnia symptoms and neurofunctional correlates among adults receiving buprenorphine for opioid use disorder

PONE-D-23-39561R1

Dear Mr. White,

We’re pleased to inform you that your manuscript has been judged scientifically suitable for publication and will be formally accepted for publication once it meets all outstanding technical requirements.

Kind regards,

Sungwoo Lim, DrPH

Academic Editor

PLOS ONE

Additional Editor Comments (optional):

Reviewers' comments:

Reviewer's Responses to Questions

**Comments to the Author**

1. If the authors have adequately addressed your comments raised in a previous round of review and you feel that this manuscript is now acceptable for publication, you may indicate that here to bypass the “Comments to the Author” section, enter your conflict of interest statement in the “Confidential to Editor” section, and submit your "Accept" recommendation.

Reviewer #1: All comments have been addressed

2. Is the manuscript technically sound, and do the data support the conclusions?

Reviewer #1: (No Response)

3. Has the statistical analysis been performed appropriately and rigorously? 

Reviewer #1: (No Response)

4. Have the authors made all data underlying the findings in their manuscript fully available?

Reviewer #1: (No Response)

5. Is the manuscript presented in an intelligible fashion and written in standard English?

Reviewer #1: (No Response)

6. Review Comments to the Author

Reviewer #1: (No Response)

7. PLOS authors have the option to publish the peer review history of their article (what does this mean?). If published, this will include your full peer review and any attached files.

Reviewer #1: **Yes: **Allison K. Wilkerson

---

## [Editor Report · Acceptance letter]

23 May 2024

PONE-D-23-39561R1 

PLOS ONE

Dear Dr. White, 

I'm pleased to inform you that your manuscript has been deemed suitable for publication in PLOS ONE. Congratulations! Your manuscript is now being handed over to our production team.

Kind regards, 

on behalf of

Dr. Sungwoo Lim 

Academic Editor

PLOS ONE